# Enhancing reactivity of SiO⁺ ions by controlled excitation to extreme rotational states

Sruthi Venkataramanababu [1,2], Anyang Li [3] ✉, Ivan O. Antonov[4], James B. Dragan[2], Patrick R. Stollenwerk [5], Hua Guo[6] & Brian C. Odom [2] ✉

Optical pumping of molecules provides unique opportunities for control of chemical reactions at a wide range of rotational energies. This work reports a chemical reaction with extreme rotational excitation of a reactant and its kinetic characterization. We investigate the chemical reactivity for the hydrogen abstraction reaction $SiO^+ + H_2 \rightarrow SiOH^+ + H$ in an ion trap. The $SiO^+$ cations are prepared in a narrow rotational state distribution, including super-rotor states with rotational quantum number ($j$) as high as 170, using a broadband optical pumping method. We show that the super-rotor states of $SiO^+$ substantially enhance the reaction rate, a trend reproduced by complementary theoretical studies. We reveal the mechanism for the rotational enhancement of the reactivity to be a strong coupling of the $SiO^+$ rotational mode with the reaction coordinate at the transition state on the dominant dynamical pathway.

A long-standing objective in chemistry is to control chemical reactions at the quantum level. One time-honored approach is by manipulating the collision energy (i.e., heating/cooling). This can be achieved with molecular beam techniques at collision energies ranging from cold (-1 K) to hyper-thermal (>500 K) regimes[1–4]. Recent progress in cooling atoms and molecules sheds valuable light on the quantum nature of reactivity at the single quantum state level under ultra-cold collision conditions (-nK)[5–8]. An alternative approach is to deposit energy into vibrational modes of the reactants, using for example optical pumping[9]. Such quantum state selective studies have revealed strong vibrational control in some reactions[10–12].

These investigations have led to a better understanding of how various types of energy promote (or inhibit) reactivity. Half a century ago, Polanyi proposed a set of rules for understanding the relative vibrational/translational efficacy in promoting atom-diatom reactions with different types of barriers[13]. Translational energy is more effective in overcoming a barrier that resembles the reactants (reactant-like), while vibrational excitation has a higher efficacy in surmounting a product-like barrier. These Polanyi rules have been extended to polyatomic reactions. The Sudden Vector Projection (SVP) model, for example, attributes the ability of a reactant mode in promoting a reaction to its projection onto the reaction coordinate[14,15].

Despite tremendous progress, there have only been a few experiments probing the effect of rotational excitation on reactivity, and these existing studies have typically involved only low rotational excitations[16–19]. Since rotational interval of a typical molecule is -2 orders of magnitude smaller than vibrational intervals, the impact of rotational excitation is difficult to observe. Besides, it is not entirely clear whether a fast-rotating reactant would necessarily enhance reactivity because rotation might increase the effective barrier, hence inhibiting the reaction. In this regard, super-rotor molecules which are promoted to highly excited rotational levels are particularly useful because the energy of rotation may approach or even exceed vibrational energy or even electronic energy holding the atoms together[20,21]. Super-rotors show fascinating collisional energy exchange properties, such as a strong propensity for conservation of angular momentum

[1]Applied Physics Program, Northwestern University, Evanston 60208 IL, USA. [2]Department of Physics, Northwestern University, Evanston 60208 IL, USA. [3]Key Laboratory of Synthetic and Natural Functional Molecule of the Ministry of Education, College of Chemistry and Materials Science, Northwest University, Xi'an 710127, P. R. China. [4]Lebedev Physical Institute, Samara 443011, Russian Federation. [5]Argonne National Laboratory, Lemont 60439 IL, USA. [6]Department of Chemistry and Chemical Biology, University of New Mexico, Albuquerque 87131 NM, USA. ✉e-mail: liay@nwu.edu.cn; b-odom@northwestern.edu

(both magnitude and direction)[22], collective transport properties such as generation of macroscopic vortices[23], and anisotropic diffusion[24] and quasi-resonant vibration-rotation energy transfer[25,26].

Recently, super-rotors were detected in the interstellar medium where they may be produced by photo-dissociation of poly-atomic molecules[27] in warm proto-planetary nebulae[28] or in interstellar shock waves[29]. Due to the exotic and exceedingly diverse conditions, interstellar chemistry is very different from that on Earth. Very little is known about chemical reactions of super-rotors and their role in astrochemical reaction mechanisms.

The main focus of experimental studies on super-rotors to date has been on probing their physical[21,30] and collisional[31–33] properties using an optical molecular centrifuge[20]. The molecular centrifuge method relies on a coherent population transfer by stimulated Raman processes for the preparation of super-rotors which makes it universally applicable to many small molecules. Even though this technique is well-suited for studying collisions, the resulting relaxation dynamics make it less ideal for studying state-dependent chemical reactions.

While optically pumping trapped molecular ions is a somewhat limited approach with regard to its applicability to a wide class of molecules, it offers several advantages that make it particularly suitable for studying the reactions of super-rotors as discussed in this work. Some of the current authors recently demonstrated the state-preparation of trapped SiO$^+$ molecules around a target rotational state and with a narrow rotational distribution ($\Delta j = 5$)[34,35] using optical pumping. Target states can range from the ground rotational state all the way to super-rotors with 230 rotational quanta. Optical pumping is several orders of magnitude faster than any relaxational dynamics and steady-state distributions can be reached within a time scale of 1 s (Fig. 1a). Furthermore, in an ion trap, the collision rate is of the order of 1 min$^{-1}$ which is substantially smaller than re-pump rates due to the laser. Together, they allow the study of a steady-state population.

SiO is one of the few molecules that act as a maser in the interstellar medium[36,37] and was first detected in interstellar clouds[38,39] and subsequently from supernovae explosions[40,41]. The reaction of SiO$^+$ with H$_2$ is of significance because it could play an important role in the production of interstellar SiO[42]. This was previously studied by Fahey

et al. and the bi-molecular rate constant was measured to be 3.2 (1.0) × 10$^{-10}$ cm$^{-3}$ molecule$^{-1}$ s$^{-1}$[37]. We report here experimental and theoretical results on the reaction kinetics of the hydrogen abstraction reaction SiO$^+$ + H$_2$ → SiOH$^+$ + H in low rotational states of SiO$^+$ and compare it with kinetics in super-rotor states. Our experimental results indicate that rotational excitation in SiO$^+$ reactant enhances reactivity by a factor of 3. Theoretical calculations provide an insight into the mechanism for enhanced reactivity as well as reproduce the trend in rotational enhancement. Specifically, the enhancement is attributed to a rotational mode specificity related to a key transition state of the reaction. Implications of super-rotor reactions for astrochemistry are discussed.

## Results

### Reaction rates

SiO$^+$ ions were loaded into a room-temperature linear Paul trap via ablation followed by photo-ionization of SiO. At ultrahigh vacuum, H$_2$ is the background gas in the trap. Due to a large excess of H$_2$, SiO$^+$ + H$_2$ → SiOH$^+$ + H is a pseudo-first-order reaction. We use the technique of Laser Cooled Fluorescence Mass Spectrometry (LCFMS)[43,44] to measure reaction rate (see "Methods"). Given our typical Coulomb crystal size and radial trapping frequency $\Omega = 2\pi \times 240$ kHz, we expect ~10 cm$^{-1}$ of micromotion energy for the outermost ions per degree of freedom. Assuming it is equally distributed between $x$, $y$, and $z$ degrees of freedom, we can have up to 30 cm$^{-1}$, which is much less than $k_b T$ ~ 200 cm$^{-1}$ at 300 K. Therefore, energy of the background hydrogen gas, not the ion, dominates the collision energy for unpumped SiO$^+$ molecules.

The SiO$^+$ molecules were pumped into narrow rotational distributions centered at specified target states using an optical pumping setup (Supplementary Fig. 1). The steady-state distribution of SiO$^+$ and the flow of population to targeted rotational states are shown in various sub-plots in Fig. 1a-d. When SiO$^+$ molecules are in super-rotor states $j$ ~ 170, the true electronic ground state is the $A$ state (Supplementary Fig. 2b). The $X(v = 0)$ state has a very little population of SiO$^+$ (Fig. 1c), and nearly all the population of SiO$^+$ is in the $A$ state (Fig. 1d). SiO$^+$ molecules are sustained in the targeted distribution limited only by their reaction with H$_2$. At excited rotational states, the internal

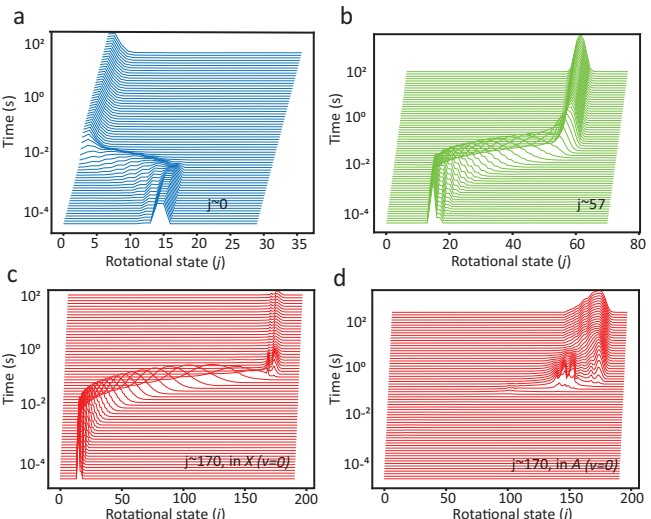

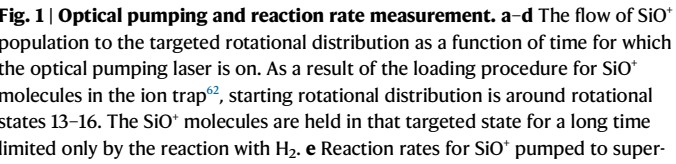

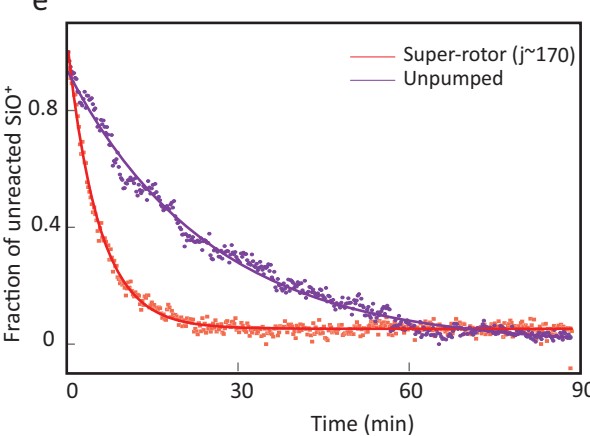

**Fig. 1 | Optical pumping and reaction rate measurement. a–d** The flow of SiO$^+$ population to the targeted rotational distribution as a function of time for which the optical pumping laser is on. As a result of the loading procedure for SiO$^+$ molecules in the ion trap[62], starting rotational distribution is around rotational states 13–16. The SiO$^+$ molecules are held in that targeted state for a long time limited only by the reaction with H$_2$. **e** Reaction rates for SiO$^+$ pumped to super-

rotor states compared to an uncontrolled rotational distribution of SiO$^+$. The experimental data points are fit (solid lines) to an exponential decay function. A higher exponential decay constant indicates that the reaction rate is faster. The reaction rate measurement was carried out at an estimated H$_2$ density of 7 (1) × 10$^6$ cm$^{-3}$.

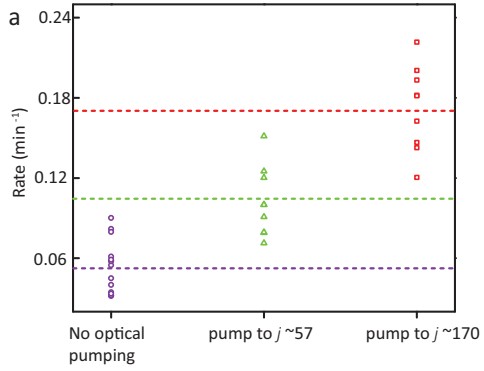
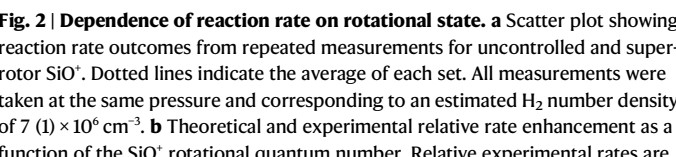
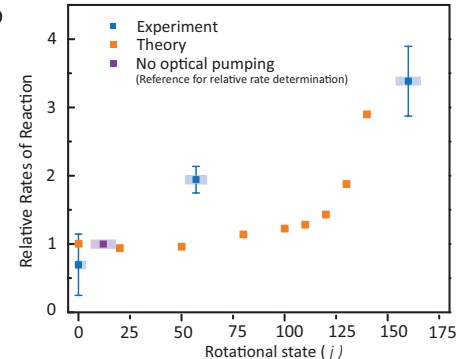

**Fig. 2 | Dependence of reaction rate on rotational state. a** Scatter plot showing reaction rate outcomes from repeated measurements for uncontrolled and super-rotor SiO⁺. Dotted lines indicate the average of each set. All measurements were taken at the same pressure and corresponding to an estimated $H_2$ number density of 7 (1) × 10⁶ cm⁻³. **b** Theoretical and experimental relative rate enhancement as a function of the SiO⁺ rotational quantum number. Relative experimental rates are calculated with respect to the case of no optical pumping (purple data point). Vertical error bars represent the $1\sigma$ statistical uncertainty. Horizontal error bars are the spread in prepared rotational distribution (with the spread encompassing more than 90% of the SiO⁺ population). Explanations for theory calculations stopping at $j=140$ are discussed in the text (see section "Reaction rates").

energy of SiO⁺ is -12$k_b$T for $j=57$ and -100$k_b$T for $j=170$ and completely dwarfs the energy of background $H_2$. Further details regarding state preparation can be found in our recent publications[34,35] and are briefly discussed in the Supplementary Information.

Figure 1e shows the fraction of unreacted SiO⁺ as a function of time at an estimated $H_2$ number density of 7 (1) × 10⁶ cm⁻³ for two cases: unpumped SiO⁺ distribution and when SiO⁺ molecules are pumped to super-rotor ($j=170$) states. The super-rotors have a larger decay rate indicating a higher rate of reaction. To investigate further, we repeated the measurement several times, during which the ion gauge was constant and well within its dynamic range, and observed rates are plotted in Fig. 2a. The average pseudo-first-order rate of reaction when the SiO⁺ is pumped to high $j$ (170) super-rotor states is 0.18 (3) min⁻¹ compared to 0.05 (2) min⁻¹ for the case of the uncontrolled SiO⁺ sample. At intermediate rotational state distribution centered at $j=57$, a rate of 0.10 (3) min⁻¹ was observed.

Theoretical enhancement in reaction rate as a function of rotational energy was calculated on the PES by a quasi-classical trajectory (QCT) method for various initial conditions. This is plotted alongside experimentally obtained results in Fig. 2b. In agreement with the experimental trend, rotational excitation of SiO⁺ enhances reactivity. Quantitatively, however, calculated rate overestimates experimental values at both low and high $j$ values. In QCT calculations, SiO⁺ reactant is treated within the rigid rotor approximation, which deteriorates for large $j$ values because of strong rotation-vibration coupling. As a result, the calculation was restricted to $j \leq 140$. Agreement between calculated and measured rate coefficients is reasonable, but not quantitative. Such levels of agreement are not uncommon for ion-molecule reactions with complex potential energy surfaces[45,46]. There are many possible theoretical reasons for the lack of quantitative agreement, chief among which is the neglect of non-adiabatic effects. Vertical excitation energy of the $X$ to $A$ state in SiO⁺ is about 0.56 eV, which suggests possible involvement of the electronically excited state in the dynamics. Our adiabatic potential energy surface contains the lowest energy regions of the $X$ and $A$ states in the reactant channel (see Supplementary Fig. 6), but non-adiabatic coupling between them is ignored. Investigating non-adiabatic effects in the dynamics of the current system is an extremely challenging task and beyond the scope of the current work. In this work, we focus on the dynamical insights provided by QCT simulations on the ground adiabatic potential energy surface to understand the observed effect and its mechanistic origin, rather than a quantitative reproduction of measured rates.

## Reaction mechanism

To understand reactivity and relative efficacy of various types of reactant excitation, a full-dimensional global adiabatic potential energy surface (PES) of the $SiOH_2^+(X^2A)$ system was constructed from high-level ab initio data. As shown in Fig. 3a, the PES has a complex topography, with multiple minima and saddle points. However, there is a dominant dynamics pathway leading to the abstraction of H by SiO⁺. This pathway, illustrated by yellow line in Fig. 3a, features a loose pre-reaction well (IM1) with a depth of merely 0.31 eV, in which the $H_2$ and SiO⁺ form a complex. This shallow well is nearly isotropically present around the SiO⁺ moiety, suggesting an electrostatic nature. The H-abstraction saddle point (TS1) is 0.12 eV below the reactant asymptote, which features a partially broken H-H bond and an incipient H-O bond. Apparently, this reaction path is barrierless with an attractive long-range interaction, although the submerged barrier serves as a reaction bottleneck for large impact parameters. Beyond the barrier, the SiOH⁺ product is formed exoergically.

In addition to this dominant reactive channel (the yellow pathway), there are several other channels leading to the two product asymptotes, represented in the figure by blue, red, and green lines. The formation of IM2 is blocked by a significant barrier in the entrance channel (TS0), as shown in Fig. 3a. Although access to IM3 from the reactants is barrierless (the A → D → E pathway marked in orange as shown in Fig. 3b), it is dynamically overwhelmed by a lower energy pathway A → B → C marked in cyan to the products. We have observed only a few trajectories leading to IM3 in our simulations, despite favorable energetics. Hence, these secondary reaction pathways are dynamically irrelevant. Most reactive trajectories show a striping mechanism in which the two reactants pass each other as the SiO⁺ picks up the H from $H_2$. On the other hand, the non-reactive trajectories are usually reflected by the repulsive potential wall in close inter-monomer distances, despite the existence of stable complexes such as IM2 and IM3, as discussed above.

In Fig. 3c, the integral cross sections are plotted for three scenarios - rotational excitation of SiO⁺, rotational excitation of $H_2$, and translational excitation. It is clear that only SiO⁺ rotation promotes the reaction, while excitation in relative translation and $H_2$ rotation inhibits the reaction. The inhibitory effect for translational mode is due apparently to the capture nature of the reaction, while that for the $H_2$ rotation can be understood by the increased effective barrier with the rotational angle[15]. To explain the enhancement in reaction due to rotation of SiO⁺, we resort to the SVP model[14]. In the sudden limit, the ability of a reactant mode to enhance the reactivity is attributed to a large projection of its normal mode onto the reaction coordinate at the

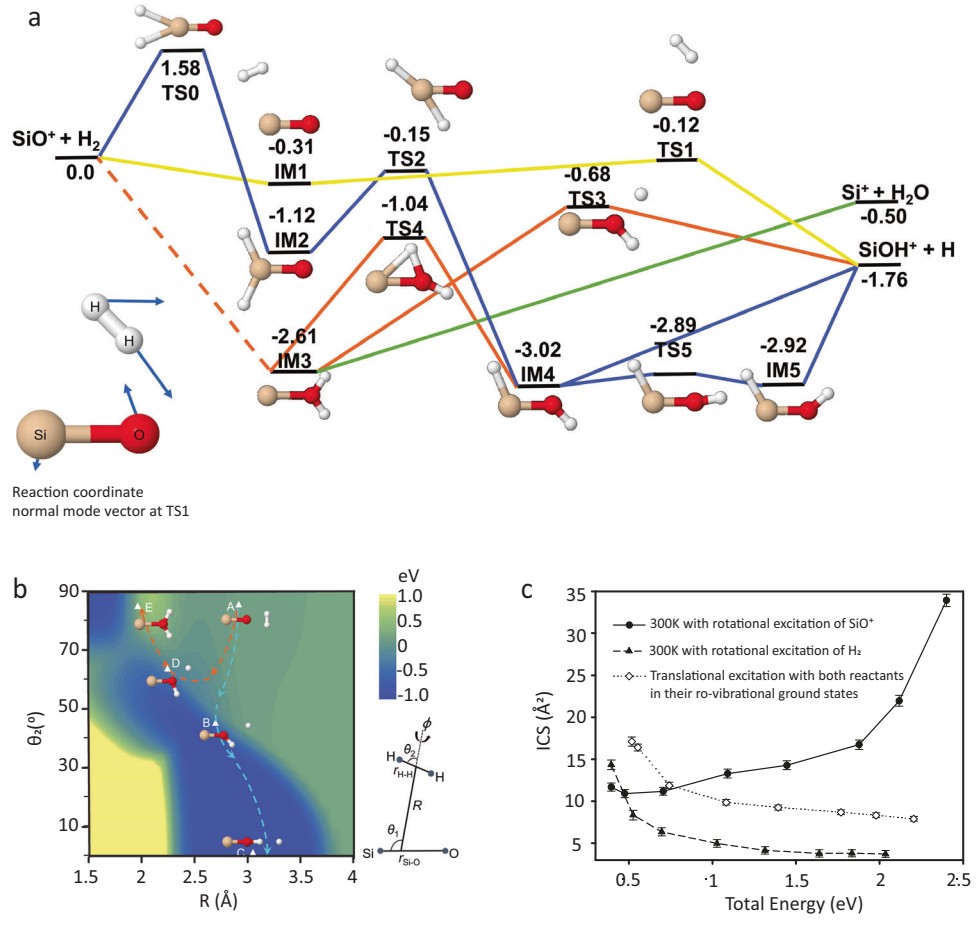

**Fig. 3 | Reaction mechanism. a** Energetics (in eV) of the ground state reaction pathways for the $SiO^+ + H_2$ reaction. Yellow line represents the dominant reaction pathway, via a submerged saddle point TS1. **b** Contours of the PES in reactant Jacobi coordinates along $R$ and $\theta_2$ with fixed $\theta_1 = 180^o$ and optimized $r_{SiO}$, $r_{HH}$, and $\phi$. **c** Comparison of integral cross sections for rotational excitation of $SiO^+$, rotational excitation of $H_2$, and translational excitation.

relevant transition state, in this case, TS1. Table 1 lists the SVP values ($0 < \zeta_i < 1$) for all the reactant modes ($i = 1$–5); evidently, the rotation of $SiO^+$ has a very large value. This implies $SiO^+$ rotation is strongly coupled with the reaction coordinate at TS1, as illustrated in the right panel of Fig. 3a, allowing facile energy flow into the reaction coordinate.

## Discussion

Optical pumping to super-rotor states provides unique opportunities for control of chemical reactions at extreme rotational energies. Preparing super-rotors with a narrow rotational distribution allows for a detailed analysis of the reaction mechanism. This work reports for the first time a chemical reaction with extreme rotational excitation of a reactant and its kinetic characterization. For the reaction $SiO^+ + H_2 \rightarrow SiOH^+ + H$, we observe an increase in the reaction rate by a factor

of 3 when the $SiO^+$ is pumped to super-rotor states as compared to the case when we do not optically pump the $SiO^+$. This rotational enhancement of reactivity is supported by QCT calculations based on the newly developed PES for this reaction. Despite the complex topography of the PES, QCT calculations suggest that a submerged barrier (TS1) has an important influence on the rotational mode specificity of the reaction. The SVP model suggests that reactivity enhancement by exciting $SiO^+$ rotation has a contribution from strong coupling with the reaction coordinate at the dominant transition state.

Extremely fast molecular rotation causes an electronic effect due to the mixing of wave functions of low-lying electronic states (such as the $X$ and $A$ states in $SiO^+$) through non-adiabatic interactions. In $SiO^+$ super-rotors, this effect is augmented by the centrifugal term that adds more energy to $X$ state than to $A$ state, resulting in closing the gap between the two states. This reverses the order of electronic states after $j = 140$ (Fig. S2) and at $j = 170$, electronic part of the wave function acquires a predominantly $A$ state character. The electronic effect complicates studying chemical reactions of super-rotors in molecules with low-lying electronic states, such as $SiO^+$, and discussions on whether electronic interactions enhance or inhibit reaction rates are beyond the scope of this paper. However, this presents a future opportunity for probing the role of non-adiabatic interactions in reactive collisions by preparing ensembles of super-rotors with well-defined rotational states, i.e., by controlling the energy gap and therefore the interaction strength between the electronic states.

**Table 1 | SVP values for rotational and vibrational modes of $SiO^+$ and $H_2$ projected onto the reaction coordinate at the dominant transition state (TS1)**

| Mode | TS1 |
| --- | --- |
| Rot ($SiO^+$) | 0.42 |
| Rot ($H_2$) | 0.08 |
| Vib ($SiO^+$) | 0.05 |
| Vib ($H_2$) | 0.12 |
| Translational | 0.15 |

The observed enhancement of reaction rate may also change our understanding of chemical reactions in interstellar media (ISM). Super-rotors are known to exist in space[27–29] where they form under the action of abundant V-UV radiation on poly-atomic molecules. At the conditions in ISM, they exist for a long time, limited by radiative decay rather than collisional relaxation processes. Therefore, they can undergo reactive collisions with other ISM species. Current astro-chemical reaction models rely largely on estimated rather than measured rates, and in most cases dependence on the internal energy levels of reactants is neglected. Reactions of super rotors are an extreme example where such a chemical intuition approach fails, and a deep understanding of underlying dynamics is necessary. The observed acceleration of exoergic barrierless $SiO^+ + H_2$ reaction is counter-intuitive; similar and more significant effects may be revealed in other chemical systems with enough energy. The importance of super-rotor chemistry in ISM is determined by the interplay between rates of production by V-UV photo-dissociation, radiative relaxation, and reactive collisions. An interesting possibility is the formation of super-rotors of non-polar symmetric molecules such as $H_2$, $O_2$, and $N_2$, which can have several orders of magnitude longer radiative lifetime compared to polar molecules[47–50]. Therefore, they may survive for a much longer time at lower densities in interstellar media (ISM) and direct observation of rotational decays for these molecules would be very unlikely. Photo-dissociation pathways leading to the formation of non-polar symmetric molecules such as $H_2$[51] or $CH_4$[52] are known, and while such objects have not been observed yet, it is plausible that they form in super-rotor states under the action of energetic V-UV quanta on poly-atomic molecules. Even in the case of polar molecules such as OH, super-rotors can be relevant in cases such as proto-stellar clouds (where the OH super-rotors were originally discovered)[28]. Gas densities on the order of $10^{12}$ cm$^{-3}$ and higher can be achieved in such environments during gravitational collapse and star formation[53]. This may be high enough for chemistry to compete with radiative cooling assuming a gas-kinetic collision rate on the order of $10^{-10}$ cm$^{-3}$ s$^{-1}$).

Our results pave the way for studies and control of chemical reactions at extreme rotational energies of the reactants, which are important for understanding the fundamental issues in reactivity as well as chemical processes in extreme environments.

## Methods
### Experimental methods
$SiO^+$ and $Ba^+$ ions were co-loaded into the ion trap via ablation followed by photo-ionization. The $Ba^+$ cloud comprised around 500–1000 $Ba^+$ ions which were continuously Doppler-cooled via transitions at 493 and 650 nm. Rotational control of $SiO^+$ was achieved via broadband rotational optical pumping (Supplementary Information).

We applied a low-voltage (0.5–1 V) excitation chirp sweeping through frequencies from 150 to 500 kHz to one of the rods in the RF trap. As the chirp hit the secular frequency of a particular species of ions in the trap, their motion was resonantly enhanced and coupled weakly to the motion of $Ba^+$. This decreased the fluorescence of $Ba^+$ ions in proportion to the number of that ion species. A photo-multiplier tube (PMT) was used to detect the fluorescent photons from the $Ba^+$ cloud. A digital counter that was binned into 1 ms intervals counted the number of PMT events in each bin. As a result, each timestamp corresponded to a particular frequency over the course of one sweep. Since secular frequencies vary from one species to another as a function of their mass, this is a convenient in situ mass spectrometry technique. Besides using LCFMS as a tool to confirm the loading of $SiO^+$ into the trap, we also used it to monitor the reaction rate of $SiO^+$ with $H_2$. Being small, $H_2$ gas molecules diffuse through the vacuum chamber walls more than any other gas and therefore it is the most difficult gas to remove from the vacuum chamber. As a result, at ultra-high vacuum levels, $H_2$ is the dominant gas in the chamber.

As $SiO^+$ reacts away to form $SiOH^+$, the dip in fluorescence shifts to lower frequencies, reflecting an increase in $SiOH^+$ concentration. At any instance in time, the fluorescence spectrum is a combination of $SiOH^+$ and the unreacted $SiO^+$. The resolving power of LCFMS technique in our ion trap is $m/\Delta m = 30$ and thus signals from $SiO^+$ (44 amu) and $SiOH^+$ (45 amu) could not be resolved. Moreover, the fluorescence of $Ba^+$ fluctuated due to drifts in frequency, on the order of a few MHz, of the lasers used to Doppler-cool $Ba^+$. On rare occasions, this also led to a loss of $SiO^+$ ions from the trap if the drift was larger than 5 MHz. To deal with the noise from fluorescence fluctuations, we used singular value decomposition (SVD) of the data to effectively isolate amplitude variations in fluorescence from the mass-dependent frequency shift of the fluorescence spectrum. With sufficient averaging, and by appropriately employing SVD analysis, we inferred the rate of reaction of $SiO^+$. Refer to Supplementary Information for details on SVD analysis.

### Potential energy surface and quasi-classical trajectory calculations
The global adiabatic PES for $SiOH_2^+(X^2A)$ is obtained from 20,147 ab initio points at the explicitly correlated unrestricted coupled cluster singles, doubles, and perturbative triples level of theory (UCCSD(T)-F12)[54] with the explicitly correlated correlation-consistent polarized core-valence triple-zeta basis set (pCVTZ-F12)[55], using MOLPRO[56]. These ab initio points were represented using the permutation invariant polynomial-neural network (PIP-NN) approach[57], which enforces the permutation symmetry between the two identical H nuclei. 17 PIPs[58] with the maximal order of 2 were used in the input layer of the NN, which has 2 layers with 30 and 60 and neurons in each layer. The NN training was performed using the Levenberg–Marquardt method[59] and early stop[60] was used to avoid over-fitting. The root-mean-square error (RMSE) for the best fit is 16 meV for the training set spanning the energy range of [−3.0, 7.4] eV, signaling a high-fidelity global fit of the ab initio data in the experimentally relevant energy range.

Quasi-classical trajectory (QCT) calculations were performed using the standard technique implemented in VENUS[61]. The trajectories were initiated with a 16.0 Å separation between reactants ($SiO^+$ and $H_2$), and terminated when products reach a separation of 8.0 Å, or when reactants are separated by 12.0 Å for non-reactive trajectories. In the $SiO^+$ rotational state-specific calculations, $SiO^+$ was treated as a rotating oscillator, with vibrational and rotational quantum numbers $v(=0)$ and $j(=0–140)$. Atomic coordinates and momenta of $H_2$ reactant were sampled randomly using a Monte Carlo approach, based on a Boltzmann distribution at 300 K. Collision energy was sampled from a Boltzmann distribution at 282 K. In the $H_2$ rotational state-specific calculations, $H_2$ was treated as a rotating oscillator with the initial state set as vibrational ground state and excited rotational states ($j = 1–14$), while $SiO^+$ was sampled randomly at 300 K. The maximal impact parameter of the collision was chosen as $b_{MAX} = 6.0$ Å, which was tested to be sufficient. The propagation time step was selected to be 0.05 fs, which allowed total energy conservation better than 1 meV for all trajectories. Batches of 10,000 trajectories have been run for each rotational state to make statistical errors all within 5%.

## Data availability
Raw data from our experiments are available via Figshare at https://doi.org/10.6084/m9.figshare.23639526.

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

## Acknowledgements
Development of dissociative analysis techniques were funded by NSF Grant No. PHY-1806861, and super rotor pumping techniques and chemistry analysis were supported by AFOSR Grant No. FA9550-17-1-0352. A.L. would like to thank the National Science Foundation of China (Grant No. 22073073) and Natural Science Basic Research Plan in Shaanxi Province of China (Grant No. 2021JM-311), and H.G. thanks the AFOSR for financial support (Grant No. FA9550-22-1-0350).

## Author contributions
S.V., I.A., and P.S. developed the laboratory techniques. S.V., J.D., I.A., and P.S. took data. B.O. led the experimental effort. A.L. and H.G. performed the theoretical calculations.

## Competing interests
The authors declare no competing interests.
