## [Peer Review File · Nature Communications]

Reviewers' comments:

Reviewer #1 (Remarks to the Author):

The manuscript reports reaction between trapped SiO⁺ molecular ions in highly excited rotational states with room temperature H₂ gas. Authors experimentally demonstrate that reaction rate increases by a factor of 3 as rotational excitation peak moves from 12-20 to 170. It is an interesting work and indeed among the first to report reactive collisions with such high rotationally excited molecules. However at this point I can not recommend publication yet. A big part of the manuscript describes reaction mechanism based on the QCT calculations. The disagreement between theory and experiment is too large in my view in order to be able to claim that one understands the effect. In fact simulation results are truncated at 140 quanta of rotational excitation, what is the reason it does not go to higher values as reached by the experiment? I think such a big discrepancy should be addressed in the manuscript and not transferred into the supplementary materials section. Another weak point is that collision partner is simply buffer gas H₂ at room temperature. The ion trapping aspect is clearly important however collisions happen at high energy.

Reviewer #2 (Remarks to the Author):

the present manuscript describes a variation in the decay time of SiO⁺ ions in a linear Paul trap, when optically pumped to very high rotational states. The variation in the decay time is used to derive the rate coefficient for the reaction of SiO⁺ with H₂, under the assumption that H₂ is the dominant gas in the vacuum chamber.

The approach of optical pumping is interesting, and has been published previously by the group. However, the setup that is used seems to be ill-equipped for reaction rate measurements. As far as I can infer from the manuscript, the H₂ gas is not introduced as a reactant gas into the trap volume. The variation in the number density that is shown in Fig 1 is just the result of fluctuations in the background pressure of the apparatus. The authors assume that 75% of the background gas is H₂. No supporting mass spectra or additional information are given. The 75% appear to be an educated guess. Given that the laser ablation targets seem to be close to the trap volume, it is possible that other background gases are responsible for the change in pressure. As the only means to determine the H₂ number density an ion gauge is mentioned. However, ion gauges have different correction factors for different trace gases, and the vacuum fluctuations may just indicate varying residual gas compositions. To remove all these uncertainties for a meaningful measurement, the H₂ should be introduced into the trap volume, and the ion gauge should be calibrated for the change in H₂ number density by an absolute pressure gauge (like a spinning-rotor gauge or a capacitance gauge).

Together with the indirect detection method (the mass resolution is not sufficient to distinguish the reactant from the product without complex data fitting), this leads to very large uncertainties. The measured rate coefficient at the initial rotational distribution [$1.3 \times 10^{-10} \text{ cm}^3\text{s}^{-1}$] is off by a factor of 2.5 compared to the previous measurement at room temperature [$3.2 \times 10^{-10} \text{ cm}^3 \text{ s}^{-1}$]. This discrepancy is almost as large as the observed rotational effect.

The comparison to the theoretical data in Figure 2b is not very convincing either. The theory curve ends at $J < 140$, while the experimental data reaches $J \sim 170$. The trend in the theory data indicates that the discrepancy at these high J numbers would be very large.

The experimental data shown here, working on residual gas fluctuations, would maybe allow for an estimate of rate coefficients (making a number of assumptions), rather than a measurement.

Concerning the relevance of the data: I think the authors overstate the significance for interstellar environments. These high-lying J states will be very short-lived (individual state lifetimes are probably on the order of milliseconds). That is why OH in high-J states can be seen in emission in space. Because the excitation decays spontaneously before a collision can occur. And densities in space are so low that collisions with H₂ may happen once a day or even once a month. An increase in the rate coefficient of a factor of 3 will not have a dramatic impact.

Reviewer #3 (Remarks to the Author):

The manuscript titled, "Rotational control of reactivity: Reaction of SiO⁺ ions in extreme rotational states" describes research that optically pumps SiO⁺ ions into high rotational states and investigates the reaction of the rotationally excited ions with H₂. The work is motivated by the existence of potentially similar reactions in interstellar media (albeit produced by a different mechanism). The work is interesting and fills a void in super rotor research—that is experimentally investigating the effect of super rotors on reaction pathways. The work is also backed up by some extensive ab initio calculations. The level of theory used for the ab initio calculations and the use of quantum scattering calculations to back up the experimental results is impressive. The work deserves to be published, however, the authors should address the following comments before the work is published:

1. What is the distribution of rotational states that the SiO⁺ ions are excited to? How many ions are excited to those high rotational states? This is briefly described in the supplemental information, but it would be helpful to know something about the initial distribution in the main text.
2. The plot in Fig. S2 suggests that other electronic states of SiO⁺ are also excited by optical pumping. Indeed, the authors acknowledge that involvement of other electronic states could contribute to the discrepancy seen in the rate constants for $j=20$ and $j=170$. What role do molecules in the other electronic states play in the reaction pathway? Are all the experimental measurements in the manuscript for molecules in the electronic ground state?
3. What about collisional relaxation of the super rotors? Do some super rotors relax before reacting with H₂?
4. What is being plotted in Fig. 1b? What is the intensity on the y-axis? The H₂ number density ($7 \times 10^6 \text{ cm}^{-3}$) should be stated on the plot or in the plot caption.
5. I don't understand the sentence from line 136-137. Wasn't this just stated in the previous sentence from 134-136?
6. Presumably the rates given in Fig. 2a are at a fixed H₂ concentration? What is the concentration used for these measurements?
7. The third experimental data point in Fig. 2b looks like it is positioned at $j = 58$ or so, but the text seems to suggest that point is at $j = 60$? Which is it? Is the point positioned in the wrong place?
8. There are data points in Fig. 2b at $j = 18$ and 20 . Are these rotational states excited at room temperature? Are the molecules pumped to these states? Why are these rotational states chosen?
9. I understand that these are challenging experiments, but it would be nice to fill in more of the rotational states in Fig. 2b? Would it be too difficult to add one or two more rotational states to that plot to get a better comparison with the theoretical plot? Especially given the size of the error bars and the discrepancy seen at $j = 170$.
10. What are the units of the color bar in Fig. 3b?

Our *response* to the reviewers' comments and questions is below:

Reviewer #1

A big part of the manuscript describes reaction mechanism based on the QCT calculations. The disagreement between theory and experiment is too large in my view in order to be able to claim that one understands the effect. In fact, simulation results are truncated at 140 quanta of rotational excitation, what is the reason it does not go to higher values as reached by the experiment? I think such a big discrepancy should be addressed in the manuscript and not transferred into the supplementary materials section.

The reviewer is correct that there are discrepancies between the calculated and measured rates. However, the level of agreement (approximately a factor of 2) is not uncommon for ion-molecule reactions with complex potential energy surfaces. There are several possible reasons for the differences, but we believe that the neglect of the nonadiabatic coupling between the ground and first excited states is probably the main culprit. To address this, the following has been included in the revised manuscript under section 2.1 : Reaction Rates (lines 156-174)

In QCT calculations, the SiO^+ reactant is treated within the rigid rotor approximation, which deteriorates for large j values because of the strong rotation-vibration coupling. As a result, the calculation was restricted to $j \leq 140$. The agreement between calculated and measured rate coefficients is reasonable, but not quantitative. Such levels of agreement are not uncommon for ion-molecule reactions with complex potential energy surfaces. Apart from experimental uncertainties, there are many possible theoretical reasons for the lack of quantitative agreement, chief among which is the neglect of nonadiabatic effects. The vertical excitation energy of the X to A state of SiO^+ is about 0.56 eV, which suggests the possible involvement of the electronically excited state in the dynamics. Our adiabatic potential energy surface contains the lowest energy regions of the X and A states in the reactant channel (see Fig. S6 in SI), but the non-adiabatic coupling between them is ignored. Investigating the non-adiabatic effects in the dynamics of the current system is an extremely challenging task and beyond the scope of the current work. In this work, we will focus on the dynamical insights provided by QCT simulations on the ground adiabatic potential energy surface to understand the observed effect and its mechanistic origin, rather than a quantitative reproduction of the measured rates.

Another weak point is that collision partner is simply buffer gas H_2 at room temperature. The ion trapping aspect is clearly important however collisions happen at high energy.

The reviewer raises a crucial point about comparison of collisional and internal state energy scales that we had not previously addressed.

For the role of ion trapping in collision energies, we remind the reviewer that the ions are sympathetically cooled by co-trapped Ba^+ into a Coulomb crystal. Given our typical crystal size and radial trapping frequency $\Omega = 2 \cdot \pi \cdot 240$ kHz, we expect $\sim 10 \text{ cm}^{-1}$ of micromotion energy for the outermost ions per degree of freedom. Assuming it is equally distributed between x, y and z, we can have up to 30 cm^{-1} , which is much less than $k_b T \approx 200 \text{ cm}^{-1}$ at 300 K. Therefore, the energy of the background hydrogen gas, not the ion,

dominates the collision energy. It should also be noted that the inner ions in the Coulomb crystal move much less and this estimate is made for the most energetic outer ions. At intermediate and high rotational states of SiO⁺, the internal energy of SiO⁺ (~12 k_BT at j=57 and ~100k_B T at j=170) dwarves the energy of the background H₂. Obviously, the trapping of the SiO⁺ ions is essential for the preparation of the super-rotor states. Furthermore, the experimental setup represents only the first step for complete control of the collision. The collision energy can, for instance, be tuned by supersonic expansion.

We have added the following brief discussion of this to section 2.1 (lines 128-138)

Given our typical crystal size and radial trapping frequency $\Omega = 2\pi \cdot 240$ kHz, we expect ~10 cm⁻¹ of micromotion energy for the outermost ions per degree of freedom. Assuming it is equally distributed between x, y, and z degrees of freedom, we can have up to 30 cm⁻¹, which is much less than k_BT = ~ 200 cm⁻¹ at 300 K. Therefore, the energy of the background hydrogen gas, not the ion, dominates the collision energy for unpumped SiO⁺ molecules. At excited rotational states, the internal energy of SiO⁺ is ~12 k_BT for j =57 and ~100k_BT for j =170 and completely dwarfs the energy of the background H₂.

Reviewer #2 (Remarks to the Author):

The variation in the number density that is shown in Fig 1 is just the result of fluctuations in the background pressure of the apparatus. The authors assume that 75% of the background gas is H₂. No supporting mass spectra or additional information are given. The 75% appear to be an educated guess. Given that the laser ablation targets seem to be close to the trap volume, it is possible that other background gases are responsible for the change in pressure. As the only means to determine the H₂ number density an ion gauge is mentioned. However, ion gauges have different correction factors for different trace gases, and the vacuum fluctuations may just indicate varying residual gas compositions. To remove all these uncertainties for a meaningful measurement, the H₂ should be introduced into the trap volume, and the ion gauge should be calibrated for the change in H₂ number density by an absolute pressure gauge (like a spinning-rotor gauge or a capacitance gauge).

Together with the indirect detection method (the mass resolution is not sufficient to distinguish the reactant from the product without complex data fitting), this leads to very large uncertainties. The measured rate coefficient at the initial rotational distribution [1.3 x 10⁻¹⁰ cm³s⁻¹] is off by a factor of 2.5 compared to the previous measurement at room temperature [3.2 x 10⁻¹⁰ cm³ s⁻¹]. This discrepancy is almost as large as the observed rotational effect. The experimental data shown here, working on residual gas fluctuations, would maybe allow for an estimate of rate coefficients (making a number of assumptions), rather than a measurement.

The measurements in Fig. 2a which report the enhancement of the reaction rates at high rotational energies of SiO⁺ were all made at a fixed ion gauge reading and well within its dynamic range. Ablation procedures to load SiO⁺ and Ba⁺ into the trap require around 100uJ and 50uJ and do not cause any significant increase in pressure in the trap. A slow reduction of pressure was observed over the span of a several months due to the pumping out of the trap by the ion pump.

Taking all these into consideration, the only relevant uncertainty for the observation of the enhancement is the statistical uncertainty shown via the scatter of individual data points in Fig. 2a. We stress that there is no plausible systematic effect between the three measurements in Fig. 2a that can explain away the statistically significant enhancement observation. The concerns with large uncertainties due to a complicated fitting procedure and fluctuating gas composition are baked into the spread of the data points in Fig. 2a. and the enhancement in reaction rate is statistically significant. Therefore, concern with the uncertainties due to knowledge of the gas composition and ion gauge calibration have no relevance to the observed enhancement.

We also understand that the uncertainty in the ion gauge and the difficulty in establishing the gas composition precludes a measurement of the rate constant of the reaction. We felt that the reviewer's suggestion was appropriate and therefore, we have moved this to the supplementary information and demoted our result to be an estimate of the rate constant for the reaction. Nevertheless, we must state that the Stern-Volmer plot is an essential result in this work because it establishes the constancy of the gas composition at different pressures. This fact is used to include the additional data point at $j=0$ in Fig. 2b, which was recorded at a slightly higher (1.6X greater) baseline vacuum pressure.

Furthermore, the fitting analysis, though involved, is corroborated by data that was taken by directly dissociating any remaining SiO^+ after a specified duration and comparing the reaction rate to that obtained by the fitting analysis. These were found to be consistent with each other. A section has been added in the supplementary information detailing this measurement. (See SI Section V: Validation of the fitting analysis)

The comparison to the theoretical data in Figure 2b is not very convincing either. The theory curve ends at $J < 140$, while the experimental data reaches $J \sim 170$. The trend in the theory data indicates that the discrepancy at these high J numbers would be very large.

See our response to Reviewer 1 above.

Concerning the relevance of the data: I think the authors overstate the significance for interstellar environments. These high-lying J states will be very short-lived (individual state lifetimes are probably on the order of milliseconds). That is why OH in high- J states can be seen in emission in space. Because the excitation decays spontaneously before a collision can occur. And densities in space are so low that collisions with H_2 may happen once a day or even once a month. An increase in the rate coefficient of a factor of 3 will not have a dramatic impact.

Rotational levels of homonuclear non-polar molecules such as H_2 , N_2 , and O_2 , have radiative lifetimes which are many orders of magnitude longer such as N_2^+ which has ro-vibrational lifetimes on the order of 1 year. Thus, homonuclear super-rotors, if produced may survive for a much longer time at lower densities in interstellar media and the direct observation of rotational decays would be very unlikely for these molecules. Furthermore, while non-polar super rotors have not been observed in space yet, photo-dissociation pathways leading to the formation of non-polar symmetric molecules, such as H_2 or CH_4 are known, and it is plausible that they form in super-rotor states under the action of energetic VUV quanta

on polyatomic molecules. Even in the case of polar molecules such as OH, super-rotors can be relevant in proto-stellar clouds (where the OH super-rotors were originally discovered). Gas densities on the order of 10^{12} cm^{-3} and higher can be achieved in such environments during gravitational collapse and star formation. This may be high enough for the super rotor reactions to compete with radiative cooling assuming gas-kinetic collision rate on the order of $10^{-10} \text{ cm}^3/\text{s}$. These are also currently emphasized in the text in the conclusions section with relevant references.

Reviewer #3 (Remarks to the Author):

1. What is the distribution of rotational states that the SiO⁺ ions are excited to? How many ions are excited to those high rotational states? This is briefly described in the supplemental information, but it would be helpful to know something about the initial distribution in the main text.

The reviewer's suggestion has been taken and Fig. 1a now shows the simulated distribution of SiO⁺ ions in the rotational states used for the kinetic measurements as well as the population flow from the initial distribution to the desired rotational states as a function of time the optical pumping laser is on. Also, it should be noted that the optical pumping laser is kept on during the entire measurement of the reaction rate. Any SiO⁺ molecule that may decay from the targeted distribution is quickly pumped back to the desired distribution by the optical pumping laser.

2. The plot in Fig. S2 suggests that other electronic states of SiO⁺ are also excited by optical pumping. Indeed, the authors acknowledge that involvement of other electronic states could contribute to the discrepancy seen in the rate constants for $j=20$ and $j=170$. What role do molecules in the other electronic states play in the reaction pathway? Are all the experimental measurements in the manuscript for molecules in the electronic ground state?

Our simulations suggest that SiO⁺ molecules spend most of the time in the lowest electronic state. Optical pumping is always carried out on the X-B transition. At extreme super-rotor energies ($j \sim 170$), the molecules are in the A state, which happens to be the true electronic ground state at those energies. As explained in SI Section II: SiO⁺ population distribution calculations via Einstein rate equations,

Near the curve crossings, the electronic states perturb each other through rotational Hamiltonian and the A-B transition, which is normally very weak, borrows intensity from the X-B transition. This effectively means that the A, $v=0$ levels near $j = 140-150$ can be pumped via A, 0 - B, 0 transition, and A, $v=1$ levels near $j = 165-175$ are pumped via the A, 1 - B, 0 transition. However, outside of these j regions A, 0 and A, 1 rotational states are essentially "dark" and therefore relax radiatively to lower j values until they either reach the perturbed region or the point where X, $v=0$ is the true ground state. In the perturbed regions, A - B excitation has a high probability of the upper state decay to the X, $v=0$ which will result in subsequent X - B pumping. Therefore, the overall process results in rotational heating of SiO⁺ molecules, populating the A state and maintaining the super-rotors rotational population through X-B pumping.

The following has also been added in the conclusions section (lines 237-251).

Extremely fast molecular rotation causes an electronic effect due to the mixing of wave functions of low-lying electronic states (such as the X and A states in SiO⁺) through non-adiabatic interactions. In SiO⁺ super-rotors, this effect is augmented by the centrifugal term that adds more energy to the X state than to the A state, resulting in closing the gap between the two states. This reverses the order of electronic states after $j=140$ (Fig. S2) and at $j=170$, the electronic part of the wave function acquires a predominantly A-state character. The electronic effect complicates studying chemical reactions of super-rotors in molecules with low-lying electronic states, such as SiO⁺, and discussions on whether the electronic interactions enhance or inhibit the reaction rates are beyond the scope of this paper. However, this presents a future opportunity for probing the role of non-adiabatic interactions in reactive collisions by preparing ensembles of super-rotors with well-defined rotational states, i.e. by controlling the energy gap and therefore the interaction strength between the electronic states.

3. What about collisional relaxation of the super rotors? Do some super rotors relax before reacting with H₂?

As the experiments are operated under UHV conditions, the collision rate is of order 1 min^{-1} and therefore insignificant compared to the optical repump rate. The final population distributions, achieved in less than 1 second, (now added in Fig. 1a to address this and other reviewer comments) are steady state. During the measurements, the pulse-shaped rotational control laser is kept on allowing the distribution to be sustained for as long as the SiO⁺ is unreacted. Thus, inelastic collisions do not play a significant role in the population distribution. Indeed, this highlights an important distinction between our method of super-rotor production and stimulated approaches which preclude directly controlling for relaxation of the population. The text has been modified to make this point clearer.

4. What is being plotted in Fig. 1b? What is the intensity on the y-axis? The H₂ number density ($7 \times 10^6 \text{ cm}^{-3}$) should be stated on the plot or in the plot caption.

The y-axis in Fig. 1b is the fraction of unreacted SiO⁺. We have added the label and the number density is also stated in the caption.

5. I don't understand the sentence from line 136-137. Wasn't this just stated in the previous sentence from 134-136?

This has been made clear in the revised text.

6. Presumably the rates given in Fig. 2a are at a fixed H₂ concentration? What is the concentration used for these measurements?

Yes, the reviewer is correct in assuming that they are at an estimated fixed H₂ concentration of $7 \times 10^6 \text{ cm}^{-3}$. We have added this in the caption as well as in the text to make it clear.

7. The third experimental data point in Fig. 2b looks like it is positioned at $j = 58$ or so, but the

text seems to suggest that point is at $j = 60$? Which is it? Is the point positioned in the wrong place?

The data point is for SiO^+ molecules in a distribution centered at $j=57$. This has been rectified and corrected for uniformity across the article.

8. There are data points in Fig. 2b at $j = 18$ and 20. Are these rotational states excited at room temperature? Are the molecules pumped to these states? Why are these rotational states chosen?

These are in fact just the nascent rotational distribution without any optical pumping. They were represented at those rotational states because the dominant SiO^+ molecules at the time of loading are in rotational states 13-16. Fig. 2b has been modified and labeled for clarity.

9. I understand that these are challenging experiments, but it would be nice to fill in more of the rotational states in Fig. 2b? Would it be too difficult to add one or two more rotational states to that plot to get a better comparison with the theoretical plot? Especially given the size of the error bars and the discrepancy seen at $j = 170$.

The authors have added an additional data point for the reaction rate of SiO^+ molecules that were cooled to the ground rotational state of SiO^+ ($j=0$) in Fig. 2b. Additional experiments for higher rotational states could not be carried out.

10. What are the units of the color bar in Fig. 3b?

The units are in eV. The figure and caption have been edited to reflect this.

REVIEWERS' COMMENTS

Reviewer #1 (Remarks to the Author):

Authors have improved the manuscript, in my opinion it is now suitable for publication in Nature Comm. This is indeed a very interesting and challenging experiment and the first to explore reactivity with extreme rotational excitations. As such it is an important contribution that will stimulate further theoretical and experimental research in this direction.

Reviewer #3 (Remarks to the Author):

My comments have been addressed and the article is acceptable for publication.

The editor has asked me to assess whether the concerns raised by Reviewer 2 have been sufficiently addressed.

The main concern raised by the reviewer concerns the measurement/estimation of the j -dependent reaction rates, which is one of the main results of the paper. The reviewer is concerned about the method for obtaining the concentrations of SiO^+ and SiOH^+ , which are indistinguishable in the LCFMS spectra and method used to extract the separate concentrations (SVD) may not do so accurately. This is indeed concerning and valid point raised by the reviewer. The authors response is to remove the concerning Stern-Volmer plot from the main text and have attempted to address the complex fitting concerns by corroborating the fitting procedure with a more direct measurement.

The reviewer is also concerned that the 300 K reaction rate does not agree with previous results. I am not sure it is valid to compare the measured reaction rate with the 300 K result. The way the molecules are prepared, the SiO^+ ions are confined to a pretty narrow ($j=13-16$) set of rotational states and is not a 300 K distribution. The authors are incorrect to make this assumption. A 300 K distribution would have population in a broader set of states. Therefore, it is not exactly relevant to compare the reaction rate for $j = 13-16$ with that of a 300 K population. Granted, I would expect the reaction rates to be similar, but not necessarily the same.

Ultimately, I agree with the authors' assessment that the most important result is the j -dependence shown in Fig. 2a, which does appear to be statistically significant.

The second concern that this reviewer had was in regards to the discrepancy between theory and experiment in Fig. 2b. This is a concern that I had as well and however upon further consideration, this result is important. It demonstrates that our current models (i.e. the rigid rotor approximation) break down at these high rotational energies and further study into these super rotor states is needed. Frankly, the discrepancy highlights the importance of this work.

The final concern of the reviewer concerns the relevance of this work to interstellar media. While, I am not an astronomer, I agree that it can be a bit of stretch to think that $j=170$ will play a critical role in interstellar reaction rates. However, the authors have provided several references to back up their claim and the importance of the work does not solely rest on the relevance to interstellar media. Ultimately, this work has provided some interesting results and could instigate further studies. The authors have sufficiently addressed the reviewers concerns and the work can be published.

Our *response* to the reviewers' comments and questions is below:

Reviewer #1

A big part of the manuscript describes reaction mechanism based on the QCT calculations. The disagreement between theory and experiment is too large in my view in order to be able to claim that one understands the effect. In fact, simulation results are truncated at 140 quanta of rotational excitation, what is the reason it does not go to higher values as reached by the experiment? I think such a big discrepancy should be addressed in the manuscript and not transferred into the supplementary materials section.

The reviewer is correct that there are discrepancies between the calculated and measured rates. However, the level of agreement (approximately a factor of 2) is not uncommon for ion-molecule reactions with complex potential energy surfaces. There are several possible reasons for the differences, but we believe that the neglect of the nonadiabatic coupling between the ground and first excited states is probably the main culprit. To address this, the following has been included in the revised manuscript under section 2.1 : Reaction Rates (lines 156-174)

In QCT calculations, the SiO^+ reactant is treated within the rigid rotor approximation, which deteriorates for large j values because of the strong rotation-vibration coupling. As a result, the calculation was restricted to $j \leq 140$. The agreement between calculated and measured rate coefficients is reasonable, but not quantitative. Such levels of agreement are not uncommon for ion-molecule reactions with complex potential energy surfaces. Apart from experimental uncertainties, there are many possible theoretical reasons for the lack of quantitative agreement, chief among which is the neglect of nonadiabatic effects. The vertical excitation energy of the X to A state of SiO^+ is about 0.56 eV, which suggests the possible involvement of the electronically excited state in the dynamics. Our adiabatic potential energy surface contains the lowest energy regions of the X and A states in the reactant channel (see Fig. S6 in SI), but the non-adiabatic coupling between them is ignored. Investigating the non-adiabatic effects in the dynamics of the current system is an extremely challenging task and beyond the scope of the current work. In this work, we will focus on the dynamical insights provided by QCT simulations on the ground adiabatic potential energy surface to understand the observed effect and its mechanistic origin, rather than a quantitative reproduction of the measured rates.

Another weak point is that collision partner is simply buffer gas H_2 at room temperature. The ion trapping aspect is clearly important however collisions happen at high energy.

The reviewer raises a crucial point about comparison of collisional and internal state energy scales that we had not previously addressed.

For the role of ion trapping in collision energies, we remind the reviewer that the ions are sympathetically cooled by co-trapped Ba^+ into a Coulomb crystal. Given our typical crystal size and radial trapping frequency $\Omega = 2 \cdot \pi \cdot 240$ kHz, we expect $\sim 10 \text{ cm}^{-1}$ of micromotion energy for the outermost ions per degree of freedom. Assuming it is equally distributed between x, y and z, we can have up to 30 cm^{-1} , which is much less than $k_b T \approx 200 \text{ cm}^{-1}$ at 300 K. Therefore, the energy of the background hydrogen gas, not the ion,

dominates the collision energy. It should also be noted that the inner ions in the Coulomb crystal move much less and this estimate is made for the most energetic outer ions. At intermediate and high rotational states of SiO⁺, the internal energy of SiO⁺ (~12 k_BT at j=57 and ~100k_B T at j=170) dwarves the energy of the background H₂. Obviously, the trapping of the SiO⁺ ions is essential for the preparation of the super-rotor states. Furthermore, the experimental setup represents only the first step for complete control of the collision. The collision energy can, for instance, be tuned by supersonic expansion.

We have added the following brief discussion of this to section 2.1 (lines 128-138)

Given our typical crystal size and radial trapping frequency $\Omega = 2\pi \times 240$ kHz, we expect ~10 cm⁻¹ of micromotion energy for the outermost ions per degree of freedom. Assuming it is equally distributed between x, y, and z degrees of freedom, we can have up to 30 cm⁻¹, which is much less than k_BT = ~ 200 cm⁻¹ at 300 K. Therefore, the energy of the background hydrogen gas, not the ion, dominates the collision energy for unpumped SiO⁺ molecules. At excited rotational states, the internal energy of SiO⁺ is ~12 k_BT for j =57 and ~100k_BT for j =170 and completely dwarfs the energy of the background H₂.

Reviewer #2 (Remarks to the Author):

The variation in the number density that is shown in Fig 1 is just the result of fluctuations in the background pressure of the apparatus. The authors assume that 75% of the background gas is H₂. No supporting mass spectra or additional information are given. The 75% appear to be an educated guess. Given that the laser ablation targets seem to be close to the trap volume, it is possible that other background gases are responsible for the change in pressure. As the only means to determine the H₂ number density an ion gauge is mentioned. However, ion gauges have different correction factors for different trace gases, and the vacuum fluctuations may just indicate varying residual gas compositions. To remove all these uncertainties for a meaningful measurement, the H₂ should be introduced into the trap volume, and the ion gauge should be calibrated for the change in H₂ number density by an absolute pressure gauge (like a spinning-rotor gauge or a capacitance gauge).

Together with the indirect detection method (the mass resolution is not sufficient to distinguish the reactant from the product without complex data fitting), this leads to very large uncertainties. The measured rate coefficient at the initial rotational distribution [1.3 x 10⁻¹⁰ cm³s⁻¹] is off by a factor of 2.5 compared to the previous measurement at room temperature [3.2 x 10⁻¹⁰ cm³ s⁻¹]. This discrepancy is almost as large as the observed rotational effect. The experimental data shown here, working on residual gas fluctuations, would maybe allow for an estimate of rate coefficients (making a number of assumptions), rather than a measurement.

The measurements in Fig. 2a which report the enhancement of the reaction rates at high rotational energies of SiO⁺ were all made at a fixed ion gauge reading and well within its dynamic range. Ablation procedures to load SiO⁺ and Ba⁺ into the trap require around 100uJ and 50uJ and do not cause any significant increase in pressure in the trap. A slow reduction of pressure was observed over the span of a several months due to the pumping out of the trap by the ion pump.

Taking all these into consideration, the only relevant uncertainty for the observation of the enhancement is the statistical uncertainty shown via the scatter of individual data points in Fig. 2a. We stress that there is no plausible systematic effect between the three measurements in Fig. 2a that can explain away the statistically significant enhancement observation. The concerns with large uncertainties due to a complicated fitting procedure and fluctuating gas composition are baked into the spread of the data points in Fig. 2a. and the enhancement in reaction rate is statistically significant. Therefore, concern with the uncertainties due to knowledge of the gas composition and ion gauge calibration have no relevance to the observed enhancement.

We also understand that the uncertainty in the ion gauge and the difficulty in establishing the gas composition precludes a measurement of the rate constant of the reaction. We felt that the reviewer's suggestion was appropriate and therefore, we have moved this to the supplementary information and demoted our result to be an estimate of the rate constant for the reaction. Nevertheless, we must state that the Stern-Volmer plot is an essential result in this work because it establishes the constancy of the gas composition at different pressures. This fact is used to include the additional data point at $j=0$ in Fig. 2b, which was recorded at a slightly higher (1.6X greater) baseline vacuum pressure.

Furthermore, the fitting analysis, though involved, is corroborated by data that was taken by directly dissociating any remaining SiO^+ after a specified duration and comparing the reaction rate to that obtained by the fitting analysis. These were found to be consistent with each other. A section has been added in the supplementary information detailing this measurement. (See SI Section V: Validation of the fitting analysis)

The comparison to the theoretical data in Figure 2b is not very convincing either. The theory curve ends at $J < 140$, while the experimental data reaches $J \sim 170$. The trend in the theory data indicates that the discrepancy at these high J numbers would be very large.

See our response to Reviewer 1 above.

Concerning the relevance of the data: I think the authors overstate the significance for interstellar environments. These high-lying J states will be very short-lived (individual state lifetimes are probably on the order of milliseconds). That is why OH in high- J states can be seen in emission in space. Because the excitation decays spontaneously before a collision can occur. And densities in space are so low that collisions with H_2 may happen once a day or even once a month. An increase in the rate coefficient of a factor of 3 will not have a dramatic impact.

Rotational levels of homonuclear non-polar molecules such as H_2 , N_2 , and O_2 , have radiative lifetimes which are many orders of magnitude longer such as N_2^+ which has ro-vibrational lifetimes on the order of 1 year. Thus, homonuclear super-rotors, if produced may survive for a much longer time at lower densities in interstellar media and the direct observation of rotational decays would be very unlikely for these molecules. Furthermore, while non-polar super rotors have not been observed in space yet, photo-dissociation pathways leading to the formation of non-polar symmetric molecules, such as H_2 or CH_4 are known, and it is plausible that they form in super-rotor states under the action of energetic VUV quanta

on polyatomic molecules. Even in the case of polar molecules such as OH, super-rotors can be relevant in proto-stellar clouds (where the OH super-rotors were originally discovered). Gas densities on the order of 10^{12} cm^{-3} and higher can be achieved in such environments during gravitational collapse and star formation. This may be high enough for the super rotor reactions to compete with radiative cooling assuming gas-kinetic collision rate on the order of $10^{-10} \text{ cm}^3/\text{s}$. These are also currently emphasized in the text in the conclusions section with relevant references.

Reviewer #3 (Remarks to the Author):

1. What is the distribution of rotational states that the SiO⁺ ions are excited to? How many ions are excited to those high rotational states? This is briefly described in the supplemental information, but it would be helpful to know something about the initial distribution in the main text.

The reviewer's suggestion has been taken and Fig. 1a now shows the simulated distribution of SiO⁺ ions in the rotational states used for the kinetic measurements as well as the population flow from the initial distribution to the desired rotational states as a function of time the optical pumping laser is on. Also, it should be noted that the optical pumping laser is kept on during the entire measurement of the reaction rate. Any SiO⁺ molecule that may decay from the targeted distribution is quickly pumped back to the desired distribution by the optical pumping laser.

2. The plot in Fig. S2 suggests that other electronic states of SiO⁺ are also excited by optical pumping. Indeed, the authors acknowledge that involvement of other electronic states could contribute to the discrepancy seen in the rate constants for $j=20$ and $j=170$. What role do molecules in the other electronic states play in the reaction pathway? Are all the experimental measurements in the manuscript for molecules in the electronic ground state?

Our simulations suggest that SiO⁺ molecules spend most of the time in the lowest electronic state. Optical pumping is always carried out on the X-B transition. At extreme super-rotor energies ($j \sim 170$), the molecules are in the A state, which happens to be the true electronic ground state at those energies. As explained in SI Section II: SiO⁺ population distribution calculations via Einstein rate equations,

Near the curve crossings, the electronic states perturb each other through rotational Hamiltonian and the A-B transition, which is normally very weak, borrows intensity from the X-B transition. This effectively means that the A, $v=0$ levels near $j = 140-150$ can be pumped via A, 0 - B, 0 transition, and A, $v=1$ levels near $j = 165-175$ are pumped via the A, 1 - B, 0 transition. However, outside of these j regions A, 0 and A, 1 rotational states are essentially "dark" and therefore relax radiatively to lower j values until they either reach the perturbed region or the point where X, $v=0$ is the true ground state. In the perturbed regions, A - B excitation has a high probability of the upper state decay to the X, $v=0$ which will result in subsequent X - B pumping. Therefore, the overall process results in rotational heating of SiO⁺ molecules, populating the A state and maintaining the super-rotors rotational population through X-B pumping.

The following has also been added in the conclusions section (lines 237-251).

Extremely fast molecular rotation causes an electronic effect due to the mixing of wave functions of low-lying electronic states (such as the X and A states in SiO⁺) through non-adiabatic interactions. In SiO⁺ super-rotors, this effect is augmented by the centrifugal term that adds more energy to the X state than to the A state, resulting in closing the gap between the two states. This reverses the order of electronic states after $j=140$ (Fig. S2) and at $j=170$, the electronic part of the wave function acquires a predominantly A-state character. The electronic effect complicates studying chemical reactions of super-rotors in molecules with low-lying electronic states, such as SiO⁺, and discussions on whether the electronic interactions enhance or inhibit the reaction rates are beyond the scope of this paper. However, this presents a future opportunity for probing the role of non-adiabatic interactions in reactive collisions by preparing ensembles of super-rotors with well-defined rotational states, i.e. by controlling the energy gap and therefore the interaction strength between the electronic states.

3. What about collisional relaxation of the super rotors? Do some super rotors relax before reacting with H₂?

As the experiments are operated under UHV conditions, the collision rate is of order 1 min^{-1} and therefore insignificant compared to the optical repump rate. The final population distributions, achieved in less than 1 second, (now added in Fig. 1a to address this and other reviewer comments) are steady state. During the measurements, the pulse-shaped rotational control laser is kept on allowing the distribution to be sustained for as long as the SiO⁺ is unreacted. Thus, inelastic collisions do not play a significant role in the population distribution. Indeed, this highlights an important distinction between our method of super-rotor production and stimulated approaches which preclude directly controlling for relaxation of the population. The text has been modified to make this point clearer.

4. What is being plotted in Fig. 1b? What is the intensity on the y-axis? The H₂ number density ($7 \times 10^6 \text{ cm}^{-3}$) should be stated on the plot or in the plot caption.

The y-axis in Fig. 1b is the fraction of unreacted SiO⁺. We have added the label and the number density is also stated in the caption.

5. I don't understand the sentence from line 136-137. Wasn't this just stated in the previous sentence from 134-136?

This has been made clear in the revised text.

6. Presumably the rates given in Fig. 2a are at a fixed H₂ concentration? What is the concentration used for these measurements?

Yes, the reviewer is correct in assuming that they are at an estimated fixed H₂ concentration of $7 \times 10^6 \text{ cm}^{-3}$. We have added this in the caption as well as in the text to make it clear.

7. The third experimental data point in Fig. 2b looks like it is positioned at $j = 58$ or so, but the

text seems to suggest that point is at $j = 60$? Which is it? Is the point positioned in the wrong place?

The data point is for SiO^+ molecules in a distribution centered at $j=57$. This has been rectified and corrected for uniformity across the article.

8. There are data points in Fig. 2b at $j = 18$ and 20. Are these rotational states excited at room temperature? Are the molecules pumped to these states? Why are these rotational states chosen?

These are in fact just the nascent rotational distribution without any optical pumping. They were represented at those rotational states because the dominant SiO^+ molecules at the time of loading are in rotational states 13-16. Fig. 2b has been modified and labeled for clarity.

9. I understand that these are challenging experiments, but it would be nice to fill in more of the rotational states in Fig. 2b? Would it be too difficult to add one or two more rotational states to that plot to get a better comparison with the theoretical plot? Especially given the size of the error bars and the discrepancy seen at $j = 170$.

The authors have added an additional data point for the reaction rate of SiO^+ molecules that were cooled to the ground rotational state of SiO^+ ($j=0$) in Fig. 2b. Additional experiments for higher rotational states could not be carried out.

10. What are the units of the color bar in Fig. 3b?

The units are in eV. The figure and caption have been edited to reflect this.

Reviewers' comments for the resubmitted version

We thank the reviewers for their time and effort in reviewing our work and for their insightful comments which have strengthened the manuscript.

Reviewer #1 (Remarks to the Author):

Authors have improved the manuscript, in my opinion it is now suitable for publication in Nature Comm. This is indeed a very interesting and challenging experiment and the first to explore reactivity with extreme rotational excitations. As such it is an important contribution that will stimulate further theoretical and experimental research in this direction.

Reviewer #3 (Remarks to the Author):

My comments have been addressed and the article is acceptable for publication.

The editor has asked me to assess whether the concerns raised by Reviewer 2 have been sufficiently addressed.

The main concern raised by the reviewer concerns the measurement/estimation of the j -dependent reaction rates, which is one of the main results of the paper. The reviewer is concerned about the method for obtaining the concentrations of SiO^+ and SiOH^+ , which are indistinguishable in the LCFMS spectra and method used to extract the separate concentrations (SVD) may not do so accurately. This is indeed concerning and valid point raised by the reviewer. The authors response is to remove the concerning Stern-Volmer plot from the main text and have attempted to address the complex fitting concerns by corroborating the fitting procedure with a more direct measurement.

The reviewer is also concerned that the 300 K reaction rate does not agree with previous results. I am not sure it is valid to compare the measured reaction rate with the 300 K result. The way the molecules are prepared, the SiO^+ ions are confined to a pretty narrow ($j=13-16$) set of rotational states and is not a 300 K distribution. The authors are incorrect to make this assumption. A 300 K distribution would have population in a broader set of states. Therefore, it is not exactly relevant to compare the reaction rate for $j = 13-16$ with that of a 300 K population. Granted, I would expect the reaction rates to be similar, but not necessarily the same. Ultimately, I agree with the authors' assessment that the most important result is the j -dependence shown in Fig. 2a, which does appear to be statistically significant.

The second concern that this reviewer had was in regards to the discrepancy between theory and experiment in Fig. 2b. This is a concern that I had as well and however upon further consideration, this result is important. It demonstrates that our current models (i.e. the rigid rotor approximation) break down at these high rotational energies and further study into these super rotor states is needed. Frankly, the discrepancy highlights the importance of this work.

The final concern of the reviewer concerns the relevance of this work to interstellar media. While, I am not an astronomer, I agree that it can be a bit of stretch to think that $j=170$ will play a critical role in interstellar reaction rates. However, the authors have provided several references to back up their claim and the importance of the work does not solely rest on the relevance to interstellar media.

Ultimately, this work has provided some interesting results and could instigate further studies. The authors have sufficiently addressed the reviewers concerns and the work can be published.